# Expanding Clinical and Genetic Landscape of *SATB2*-Associated Syndrome

**DOI:** 10.3390/genes16101229

**Published:** 2025-10-17

**Authors:** Verdiana Pullano, Federico Rondot, Ilaria Carelli, Slavica Trajkova, Silvia Carestiato, Simona Cardaropoli, Diana Carli, Elisa Biamino, Fabio Sirchia, Giuseppe Reynolds, Roberto Keller, Elena Shukarova-Angelovska, Giovanni Battista Ferrero, Alfredo Brusco, Alessandro Mussa

**Affiliations:** 1Department of Neuroscience “Rita Levi Montalcini”, University of Torino, 10126 Turin, Italy; verdiana.pullano@unito.it (V.P.); slavica.trajkova@unito.it (S.T.); silvia.carestiato@unito.it (S.C.); 2Department of Medical Sciences, University of Torino, 10126 Turin, Italy; federico.rondot@unito.it (F.R.); ilaria.carelli@unito.it (I.C.); diana.carli@unito.it (D.C.); 3Medical Genetics Unit, Città della Salute e della Scienza University Hospital, 10126 Turin, Italy; 4Department of Public Health and Pediatric Sciences, University of Torino, 10126 Turin, Italy; simona.cardaropoli@unito.it (S.C.); ebiamino@cittadellasalute.to.it (E.B.); giuseppe.reynolds@unito.it (G.R.); alessandro.mussa@unito.it (A.M.); 5Department of Molecular Medicine, University of Pavia, 27100 Pavia, Italy; fabio.sirchia@unipv.it; 6Medical Genetics Unit, IRCCS San Matteo Foundation, 27100 Pavia, Italy; 7Adult Autism Centre DSM ASL Città di Torino, 10138 Turin, Italy; roberto.keller@aslcittaditorino.it; 8Department of Endocrinology and Genetics, University Clinic for Children’s Diseases, Medical Faculty, University Sv. Kiril I Metodij, 1000 Skopje, North Macedonia; elena.shukarova.angelovska@medf.ukim.edu.mk; 9Department of Clinical and Biological Sciences, University of Torino, 10149 Orbassano, Italy; giovannibattista.ferrero@unito.it

**Keywords:** *SATB2*, Glass syndrome, neurodevelopmental disorders

## Abstract

Background: *SATB2*-associated syndrome (SAS), also known as Glass syndrome, is a neurodevelopmental disorder (NDD) characterized by intellectual disability, developmental delay, absent or limited speech, and distinctive craniofacial and dental anomalies. It is caused by autosomal dominant pathogenic variants in the *SATB2* gene, which plays a crucial role in brain, dental, and jaw development. Due to its variable phenotype, clinical diagnosis can be challenging, necessitating genetic confirmation. Methods: We present six new cases of SAS with *SATB2* germline variants identified through next generation sequencing (NGS) technologies, expanding the known genetic and clinical spectrum of the syndrome. Detailed clinical phenotyping was performed for all patients. Results: Our cohort exhibits a broad range of clinical manifestations consistent with SAS, encompassing severe intellectual disability, profound speech delay, various palatal and dental abnormalities. We report the oldest adult patient (56 years old) carrying an in-frame duplication, and a pediatric patient with a missense variant who presented a significant reduction in visual acuity, likely of neurological or cortical origin, in the absence of ophthalmological abnormalities. *SATB2* variants include three missenses, two in-frame deletion/duplication and one frameshift variant, several of which are novel and classified as likely pathogenic or pathogenic according to ACMG guidelines. Conclusions: This report provides new clinical and genetic insights into the landscape of SAS. Our findings confirm the phenotypic heterogeneity of SAS and highlight the critical role of comprehensive genetic testing for accurate diagnosis in NDD patients.

## 1. Introduction

*SATB2*-associated syndrome (SAS; OMIM #612313), also called Glass syndrome, is a rare multisystemic disorder caused by haploinsufficiency or dysfunction of the *SATB2* gene (OMIM #608148), located on chromosome 2q33.1. Clinically, SAS is primarily characterized by neurodevelopmental impairment with severely limited or absent expressive speech, intellectual disability, and dental anomalies. Additional recurrent features include craniofacial dysmorphisms, palatal defects, variable behavioral anomalies, growth retardation, low bone density, and epilepsy [1].

The most distinctive hallmark of SAS is the early and persistent deficit in verbal communication, with nearly all affected individuals presenting delayed language milestones, often progressing to absent or extremely limited speech [1,2]. Orofacial abnormalities, mostly represented by dental anomalies and cleft palate, are reported in up to 50% of cases and have been associated with more severe language and feeding difficulties [2]. In addition, a failure to thrive, associated with normal linear growth, is frequently observed in individuals with SAS, often accompanied by scoliosis and low bone density [2,3].

The *SATB2* gene encodes a nuclear matrix-associated transcription factor that organizes higher-order chromatin architecture and regulates transcriptional programs essential for neurodevelopment, craniofacial morphogenesis, and osteogenesis [4]. SATB2 binds matrix-attachment regions (MARs) of DNA and recruits chromatin-modifying enzymes, including histone deacetylases (HDACs) and acetyltransferases (HATs), to control gene activation or repression at key regulatory elements [5]. During corticogenesis, SATB2 represses Ctip2/BCL11B to specify upper-layer cortical projection neurons, thereby orchestrating callosal connectivity [6,7]. In skeletal development, SATB2 functions as a molecular hub by directly binding enhancers of osteogenic genes such as *Hoxa2*, *IBSP*, and *BGLAP*, and synergistically interacting with ATF4 and RUNX2 to potentiate osteoblast differentiation [8]. Mice with combined Satb2 and Runx2 or Atf4 deficiency exhibit severe skeletal defects, underscoring SATB2 pivotal role in bone formation [4,9].

Genotype–phenotype correlation studies suggest that variant position within *SATB2* domains influences clinical severity. Missense variants located outside the main functional domains are associated with lower scores in impaired cognition, behavior, sleep, and sialorrhea categories, reflecting milder neurodevelopmental impairment.

To better systematize this phenotypic variability, Zarate et al. [10] developed a standardized SAS severity scoring system, enabling structured evaluation of neurodevelopmental and systemic involvement across 164 genetically confirmed cases. This tool revealed that null variants and large chromosomal deletions are associated with a more severe phenotype, especially in terms of communication, feeding, and adaptive functioning.

Although knowledge of SAS has expanded, its diagnosis during early childhood remains challenging, especially when craniofacial dysmorphisms are subtle or absent. The present study contributes to the clinical delineation of SAS by reporting a new case series of molecularly confirmed individuals, with detailed phenotypic documentation. Our aim is to enrich current understanding of the clinical spectrum, particularly in relation to neurodevelopment, craniofacial traits, and familial background, and to contribute to future genotype–phenotype stratification efforts.

## 2. Materials and Methods

### 2.1. Human Subjects

Subjects were recruited and examined by a network of collaborating clinical geneticists and pediatricians. Genetic testing primarily utilized trio exome sequencing. Exceptions included Patient 1, who underwent singleton exome sequencing, and Patient 6 who underwent genome sequencing. Sanger sequencing for variant confirmation was conducted at external laboratories. The reported data were derived from molecular diagnostic reports and clinical documentation.

Preliminary classification of variants was conducted using the Franklin by Genoox germline variant interpretation platform (franklin.genoox.com), which applies the American College of Medical Genetics and Genomics (ACMG) criteria [11]. For cases with confirmed parental segregation, the PS2 criterion (de novo occurrence) was manually added. Variant allele frequencies were assessed using data from the Genome Aggregation Database (gnomAD v4.1). Bioinformatic prediction for missense variants was evaluated using MetaRNN (http://www.liulab.science/metarnn.html), a pathogenicity prediction score for human nonsynonymous SNVs (nsSNVs). It combined data from 28 high-level annotation scores, comprising 16 functional prediction scores (like SIFT, PolyPhen2_HDIV, PolyPhen2_HVAR, MutationAssessor, PROVEAN, VEST4, M-CAP, REVEL, MutPred, MVP, PrimateAI, DEOGEN2, CADD, fathmm-XF, Eigen, and GenoCanyon), 8 conservation scores (GERP, phyloP100way_vertebrate, phyloP30way_mammalian, phyloP17way_primate, phastCons100way_vertebrate, phastCons30way_mammalian, phastCons17way_primate, and SiPhy), and 4 allele frequency datasets (1000 Genomes Project, ExAC, gnomAD exome, and gnomAD genome). These inputs are integrated into an ensemble prediction framework based on a deep recurrent neural network (RNN), which outputs the probability of a non-synonymous single nucleotide variant (nsSNV) being pathogenic. The probability is expressed as a score ranging from 0 to 1, with a cut-off of 0.5 applied for tolerance prediction [12]. In accordance with a recently proposed calibration strategy [13] we adopted the following thresholds to map MetaRNN scores to evidence strengths (≥0.748: supporting evidence of pathogenicity, ≥0.841: moderate evidence of pathogenicity, ≥0.939: strong evidence of pathogenicity). However, in line with ACMG/AMP guidelines, all in silico predictors were consistently considered as supportive but non-conclusive evidence. No variant was classified solely on the basis of predictor scores.

Protein domain structures and *SATB2* variant locations were visualized using the Illustrator for Biological Sequences (IBS) tool [14]. Domain annotations were retrieved from Ensembl protein domain Pfam for transcript NM_001172509.2 (hg38).

### 2.2. Modeling Pipeline and Statistics

Wild-type (WT) structures were retrieved from the AlphaFold Protein Structure Database [15], and variant backbones were modeled with the AlphaFold server from the corresponding mutated amino-acid sequences. Resulting mmCIFs were converted to PDB and pre-optimized with FoldX v5 (RepairPDB, 3 consecutive passes per model). We then ran Stability independently on WT and mutant models. For each variant, ΔΔG was defined as TotalEnergy(Mutant) − TotalEnergy(WT) (kcal/mol; ΔΔG > 0 = destabilizing; ΔΔG < 0 = stabilizing). To account for model uncertainty, ΔΔG was computed across the five AlphaFold ranked models per genotype (ranked_0–ranked_4) and summarized as mean ± SD with 95% confidence intervals. Per-residue confidence (pLDDT) was read from the B-factor field of AlphaFold PDBs; for the affected region (SATB2 HOX domain residues 616–671, ±10 residues) we report the per-residue values and their mean ± SD (Appendix A), and we interpret geometry primarily where pLDDT ≥ 70. Local WT–mutant superpositions were performed in UCSF ChimeraX [v1.10.1] over Cα atoms within the 616–671 residue window around the altered site. Structures were visualized and superposed in UCSF ChimeraX for qualitative inspection (local RMSD, hydrogen-bond networks, and packing contacts) and figure preparation [16]. AlphaFold confidence limits (pLDDT/PAE) and the empirical nature of FoldX imply typical uncertainties. Accordingly, ΔΔG values are treated as qualitative indicators until supported by functional assays.

## 3. Results

### 3.1. Patients Descriptions (Table 1)

Patient 1 (Figure 1A): The patient is a male who was referred to our unit at the age of 56 yrs. for a pervasive NeuroDevelopmental Disorder (NDD) characterized by absent speech, autistic behavior, and moderate intellectual disability. The NDD was related to an episode of infectious encephalitis, although the etiology was never clearly established. This occurred in conjunction with a single febrile seizure. Brain Magnetic Resonance Imaging (MRI) was largely unremarkable, except for an isolated focal area of gliosis of nonspecific significance located in the subcortical white matter of the right superior frontal gyrus. Neuropsychiatric evaluation revealed good receptive verbal comprehension, use of gestural communication, sleep disturbances, and ritualistic, stereotyped, and compulsive behaviors, including self-directed aggression. A significant impairment was observed across all domains of adaptive functioning. In recent years, episodes of fluctuating lower limb hyposthenia have been reported. The patient also exhibits peculiar eating behavior, including hyperphagia with a BMI of 20 (stature of 175 cm and weight of 61 kg). Dental anomalies could not be fully characterized due to multiple prior dental extractions, which have resulted in partial edentulism with only the incisors remaining. The patient also presents with bilateral pes planus and brachydactyly with an enlargement of the distal phalanx of the thumb. Genetic testing included array-CGH, which yielded non-diagnostic results. Singleton exome sequencing (parental samples were unavailable) subsequently identified a heterozygous in-frame duplication of two amino acids in the *SATB2* gene (NM_001172509.2): c.961_966dup, p.(Ile321_Ala322dup).

Patient 2 (Figure 1B): This patient was a male born at term (40 weeks of gestation). Neonatal auxometric parameters were within normal limits; the Apgar score was 9 at the first minute. In the first days of life, facial dysmorphisms were noted, including a mildly flattened nasal bridge, subtle midface hypoplasia, laterally sparse eyebrows, and syndactyly of the proximal phalanges of the second and third toes. At 19 months of age, the patient was diagnosed with laryngomalacia, which required endoscopic correction. Independent walking was achieved at 6 years of age and was described as markedly delayed and unstable, with a persistently flexed posture. At the most recent clinical evaluation (10 years of age), anthropometric measurements were: weight 25 kg (5th centile), height 127 cm (4th centile), and head circumference 52 cm (27th centile). Brain MRI revealed diffuse white matter demyelination, ventricular enlargement, and global white matter volume reduction. Ophthalmologic evaluation documented reduced visual acuity, likely of neurological (cortical) origin, in the absence of ocular abnormalities. Genetic work-up included array-CGH, which was non-diagnostic. Trio exome sequencing subsequently identified a de novo heterozygous missense variant in the *SATB2* gene (NM_001172509.2): c.1573G>A, resulting in the amino acid substitution p.(Glu525Lys).

Patient 3 (Figure 1C): A 9-year-old female was referred for global NDD and facial dysmorphisms, with an unremarkable family history. The pregnancy was reported as uneventful, with delivery occurring at 40 + 0 weeks of gestation. Birth parameters included a weight of 3790 g and a length of 51 cm, OFC was not reported. Physical examination revealed multiple craniofacial anomalies, including cleft palate, unruly dentition, short philtrum, small mouth opening, under folded helices, hypertelorism, downward slanting palpebral fissures, blue sclerae, long eyebrows, and prominent ears. Independent walking was achieved at 32 months. A clinical evaluation at 4 years of age confirmed persistent motor developmental delay. At 9 years of age, the patient was able to articulate only a few words, but presented non-verbal communication abilities, eye contact, and understanding spoken language. Cognitive assessment yielded an IQ score of 50, consistent with moderate intellectual disability. Brain MRI was unremarkable apart from the presence of arachnoid cysts. No seizures have been observed to date, although electroencephalogram (EEG) recordings demonstrated generalized immaturity of background activity. Array-CGH was normal, while a trio exome sequencing identified a de novo heterozygous variant in the *SATB2* gene (NM_001172509.2): c.1196G>A, resulting in the missense change p.(Arg399His).

Patient 4: This patient was referred for global NDD in the context of an unremarkable family history. No complications were reported during pregnancy or the perinatal period. Physical examination revealed nonspecific facial dysmorphisms and an abnormal bilateral morphology of the distal phalanges of the thumbs. Neuropsychiatric evaluation highlighted global neurodevelopmental and language immaturity, with findings consistent with intellectual disability and a clinical diagnosis of autism spectrum disorder (ASD). Behavioral assessment identified obsessive–compulsive traits and episodes of aggressive behavior. The patient experienced transient epileptic events, and EEG recordings confirmed abnormal cerebral activity. The array-CGH yielded inconclusive results. Subsequent trio exome sequencing revealed a de novo heterozygous missense variant in the *SATB2* gene (NM_001172509.2): c.587C>T, resulting in the amino acid substitution p.(Pro196Leu).

Patient 5: This female patient was referred at 2 years old for a significant speech delay and absent social interaction. Family history was negative, and she had no perinatal complications. Upon physical examination, she presented with a rounded facial shape, prominent nasal bridge, and micrognathia. Her developmental milestones were substantially delayed: independent walking was achieved at 24 months, and her first spoken words were not reported before 48 months, indicating a significant delay in expressive language development. A neuropsychiatric evaluation conducted at 9 years of age revealed an unstable gait and marked deficits in both gross and fine motor skills. Her expressive speech remained severely limited, and she continued to exhibit absent social engagement alongside persistent sleep disturbances. Cognitive assessment documented an IQ score of 60, consistent with moderate intellectual disability. An EEG showed an abnormal pattern characterized by alpha-theta rhythm with spikes and slow waves, as well as a degenerated spike–wave complex in the frontal regions. After a normal array-CGH, a trio exome sequencing identified a de novo heterozygous pathogenic variant in the *SATB2* gene (NM_001172509.2): c.857dup, resulting in a frameshift and premature termination variant p.(Pro287AlafsTer17).

Patient 6: This 11-year-old female was born after an uneventful pregnancy and delivery. At birth, she presented with a cleft palate, lingualized lower incisors, and congenital clubfoot. At 11 years old, growth assessment showed she was underweight (25 kg, −2.2 SD), with height and head circumference at the lower limits of normal. Neurodevelopmental evaluation revealed intellectual disability and language delay, evident from early childhood. An ophthalmological assessment identified strabismus and astigmatism. Brain MRI showed hypoplasia of the corpus callosum. EEG revealed abnormalities in the absence of clinical seizures. Karyotype and array-CGH were normal. Trio exome sequencing identified a de novo in-frame deletion in the *SATB2* gene (NM_001172509.2): c.1979_1981del, resulting in the deletion of one isoleucine residue p.(Ile660del). Family history was notable for a maternal uncle with a cleft lip and palate.

**Table 1 genes-16-01229-t001:** Clinical summary of six patients with Glass syndrome.

Case	NDD	ASD	Language Delay	MRI Anomalies	Behav. Anom.	Motor Delay	Sleep Difficult.	Facial Dysmorphism	Cleft Palate	Dental Anomalies	Growth Retardation	Clinical Seizures
1	+	+	+	−	+	+	+	+	−	+	−	−
2	+	+	+	+	+	+	−	+	−	−	+	−
3	+	−	+	−	−	+	−	+	+	+	−	−
4	+	+	+	NA	+	+	−	+	−	−	−	+
5	+	+	+	NA	+	+	+	+	−	−	−	−
6	+	−	+	+	−	+	−	+	+	+	+	−

Notes: Behav. Anom.: behavioral anomalies; NA: not available; +: present; −: absent. Extended schematic representation of genotype–phenotype features of *SATB2* patients including SAS severity scores calculation are available in Appendix A.

### 3.2. Variants Analysis

The *SATB2* gene (OMIM #608148), located on chromosome 2q33, spans approximately 191 kb and comprises 12 exons. It encodes a 733-amino-acid DNA-binding protein that selectively binds AT-rich sequences and is highly conserved across species [17,18]. According to the Genome Aggregation Database (gnomAD v4.1; https://gnomad.broadinstitute.org/; accessed on 1 August 2025), *SATB2* exhibits a high probability of loss-of-function intolerance (pLI = 1.0) and a Z-score of 5.53, supporting its classification as a haploinsufficient gene with strong constraint against variation.

Table 2 provides a detailed summary of the identified variants in our cases, which are also schematically summarized on the protein in Figure 2A. Extended data on ACMG criteria and variant classification, together with Clinvar variant information, are reported in Appendix A.

#### 3.2.1. Missense Variants

Cases 2–4 harbor de novo heterozygous missense variants: c.1573G>A p.(Glu525Lys) within the CUT2 domain, c.1196G>A p.(Arg399His) within the CUT1 domain and c.587C>T p.(Pro196Leu) within the CUTL domain, respectively, as illustrated in Figure 2A. All variants were absent from the GnomAD population database. Scores derived from MetaRNN (http://www.liulab.science/metarnn.html; accessed on 1 October 2025) demonstrate a supportive but non-conclusive pathogenicity assessment of all three variants.

MetaDome analysis (https://stuart.radboudumc.nl/metadome/; accessed on 1 October 2025) confirms they are in regions intolerant to variation, further supporting their pathogenicity (Figure 2B). Based on ACMG (Table 2) criteria, all three variants were classified as likely pathogenic.

The c.1196G>A p.(Arg399His) variant has been previously reported in ClinVar (ID 373069), the LOVD database (ID SATB2_000007), and described by [19]. The c.1573G>A p.(Glu525Lys) variant has also been reported in ClinVar (ID 2577951). To the best of our knowledge, the c.587C>T p.(Pro196Leu) variant has not been previously reported (Table 2).

#### 3.2.2. Indel Variants

The remaining three cases carry two in-frame variants (one duplication and one deletion) and one frameshift duplication, all of which are absent in the control population (gnomAD v4.1). Based on ACMG criteria [11], all variants were classified as likely pathogenic or pathogenic (Table 2).

MetaDome analysis (https://stuart.radboudumc.nl/metadome/; accessed on 1 October 2025) indicates that the regions affected by the in-frame indel variants fall within protein segments that are intolerant to variation, further supporting the potential pathogenicity of these alterations (Figure 2B).

The c.857dup p.(Pro287AlafsTer17) variant introduces a frameshift, predicted to result in nonsense mediated mRNA decay, and it has not been previously described.

The c.1979_1981del p.(Ile660del) variant results in an in-frame deletion affecting a conserved residue and has been previously reported in ClinVar as pathogenic (ID 3062110). Exploiting in silico 3D protein stability prediction analysis, we generated and compared the WT and mutated (p.(Ile660del)) form of the SATB2 HOX domain (Figure 3). The prediction indicates a potential increase in protein stability (ΔΔG = −41.02 ± 18 kcal/mol; Appendix A). Nevertheless, these results were considered only as qualitative evidence and will require further functional assays to substantiate the interpretation of the variant’s effect (Figure 3).

The c.961_966dup p.(Ile321_Ala322dup) variant causes an in-frame insertion involving two conserved residues and has not been previously reported. In this case, parental segregation analysis was not performed, so the de novo status could not be confirmed (Table 2).

## 4. Discussion

*SATB2*-associated syndrome (SAS) is primarily characterized by severe speech impairment, intellectual disability, behavioral issues, palatal defects, and dental anomalies. Despite these features, a definitive clinical diagnosis is often difficult to obtain because the variable and nonspecific signs overlap with other NDDs. The diagnostic challenge is further compounded by the fact that nearly all cases result from de novo variants [1], rendering family history uninformative. Consequently, diagnosis is frequently delayed, and it is most often confirmed through next-generation sequencing (NGS).

This manuscript reports on six novel cases of SAS, highlighting key clinical and genetic insights. We present the oldest described case of SAS in a 56-year-old male (patient 1), who demonstrates comparable age-dependent non-progressive clinical features previously described in the literature [20]. This patient’s phenotype has been stable and non-progressive throughout his life. The only new feature to emerge was a late-onset, mild hyposthenia of the lower limbs at age 45, which appeared to be related to episodes of compulsive activity. This patient also exhibited the notable combination of hyperphagia and a normal-to-low BMI, a feature that aligns with recent in vitro evidence suggesting SATB2 dysregulation may impact the metabolism of carbohydrates and amino acids [21,22]. The patient’s delayed diagnosis at age 56 yr., previously misattributed to an unclarified post-infective event, underscores the critical importance of updated clinical evaluations to avoid misdiagnosis and improve patient management. Genetically, this patient carries a heterozygous in-frame duplication of two amino acids p.(Ile321_Ala322dup), affecting a highly conserved linker domain, which is classified as likely pathogenic (LP). This variant is not found in the GnomAD database, but its de novo status could not be confirmed due to parental DNA unavailability.

Our findings from two additional cases, patient 2 and patient 3, corroborate existing literature on the impact of missense variants in critical functional domains of SATB2. These patients carry de novo missense variants, p.(Arg399His) and p.(Glu525Lys), affecting the CUT1 and CUT2 domains, respectively. The CUT1 and CUT2 domains of SATB2 play a central role in its ability to function as a global chromatin organizer. These two highly conserved DNA-binding motifs act in coordination with the C-terminal homeodomain to recognize and anchor SATB2 to matrix-attachment regions (MARs), thereby establishing the chromatin loops necessary for long-range transcriptional regulation [1]. Functional studies suggest that the CUT1 domain is particularly important for initiating the interaction with chromatin, whereas CUT2 facilitates the release of SATB2 once transcriptional remodeling has been executed, ensuring the dynamic nature of chromatin binding [23]. Through this cycle of attachment and dissociation, SATB2 maintains a finely tuned balance between chromatin stability and plasticity, which is essential for the regulation of gene networks involved in cortical neuron specification, craniofacial patterning, and odontogenesis [4,24].

The clinical presentation of these two patients is remarkably severe, with a complete absence of expressive language accompanied by global motor and neurodevelopmental delay, together with variable craniofacial, skeletal, and dental anomalies. Such features reinforce the concept that specific *SATB2* variants can profoundly disrupt protein function. Previous genotype–phenotype studies have already emphasized the severe presentation in patients with CUT1 and CUT2 domains to pathogenic changes [10].

We also identified novel variants that expand our understanding of the gene’s functional domains. Patient 4 carries the p.(Pro196Leu) variant, a novel missense change in the CUT-Like (CUTL) domain. While this domain is highly conserved, its functional role in SATB2, particularly in DNA binding, remains largely unexplored [25]. However, structural and biochemical work on SATB1 demonstrated that the ULD–CUTL tandem is essential for the multiple-domain–coordinated mechanism of DNA recognition, with CUTL contributing, beyond CUT1, CUT2 and HOX, to the DNA binding capacity of full-length protein and enabling simultaneous engagement of two DNA targets, a plausible basis for MAR-anchored loop formation. The high degree of homology (86%) between the CUTL domains of SATB1 and SATB2, reaching complete conservation at the position of the missense variant p.(Pro196Leu) (Figure 4A,B), suggests a potential effect on ULD–CUTL coupling and oligomerization, ultimately leading to reduced multi-site DNA engagement and defective chromatin-loop organization as a potential pathogenic mechanism [25].

Variant p.(Ile660del) was found in patient 6, affecting the HOX domain. Recently, functional assays have shown that missense variants within this domain impair transcriptional repression and disrupt normal nuclear localization, resembling the effects observed in HOX-domain-lacking truncating variants and supporting its critical role in MAR (Matrix Attachment Region) binding and chromatin remodeling activity. It is therefore conceivable that the variant p.(Ile660del) has a similar role in the loss of functional integrity of the HOX domain [26]. Our 3D folding stability predictions suggest increased stability of the mutant protein compared with the wild type. While this finding may imply an effect on protein function, it should be interpreted with caution and confirmed through dedicated functional assays to elucidate the underlying pathogenic mechanism (Figure 3). This represents the first reported in-frame deletion in this domain and indeed in any region of the *SATB2* gene.

Furthermore, in patients 2 and 6, neuroradiological anomalies were reported as diffuse white matter hypomyelination and volume loss for patient 2 and hypoplasia of the corpus callosum for patient 6. Those findings are consistent with previous reports that describe abnormality of white matter, delay of myelination, ventriculomegaly and small corpus callosum as the four most frequently reported neuroimaging anomalies [19].

Interestingly, patient 2 presented with a unique visual phenotype of severely reduced visual acuity without apparent ocular abnormalities combined with neuroradiological features of diffuse white matter hypomyelination and volume loss, suggesting a cortical origin of the vision loss. It is the first time, to our knowledge, that a central vision impairment is reported in a SAS patient. According to the Human Protein Atlas, SATB2 shows strong nuclear expression in pyramidal neurons of the cerebral cortex, including the occipital lobe and visual areas, supporting its potential role in the development and function of cortical circuits essential for the processing of visual information, combined with the assessed role of SATB2 in the regulation of the callosal projection during cerebral development [6].

A consistent finding across our cohort is a trend toward lower-end body weight in patients 1, 2, and 6. Detailed longitudinal growth curves were not available for our patients; however, the most recent clinical measurements and parental reports consistently indicated a history of chronic low weight. This finding is in line with previous observations in individuals with *SATB2*-associated syndrome (SAS), where growth delay predominantly manifests as reduced weight-for-length or BMI, reflecting difficulties in sustaining adequate weight gain over time [3]. Interestingly, the role of SATB2 as a key regulator for gene networks involved in growth and energy metabolism was suggested by recent in vitro studies that highlight a defective use of several energy substrates of cellular metabolism [21,22]. Further investigation using patient-derived cellular models will be essential to determine whether this metabolic dysregulation is a direct contributor to the SAS phenotype.

## 5. Conclusions

This case series expands the clinical and mutational spectrum of *SATB2*-associated syndrome (SAS) by describing six new patients carrying rare or novel *SATB2* variants, including missense, in-frame, and frameshift changes. Our findings confirm the marked phenotypic variability of SAS and emphasize the importance of comprehensive genetic testing for individuals with unexplained neurodevelopmental delay, particularly when accompanied by speech impairment and craniofacial anomalies. Notably, we report the oldest, molecularly confirmed case of SAS to date, as well as a unique visual phenotype likely of cortical origin, further broadening the phenotypic landscape. In addition, the recurrent observation of growth parameters at the lower limits of normal and recent evidence of metabolic dysregulation invite future studies to explore SATB2’s role in systemic homeostasis. Altogether, our results underscore the need for integrated clinical, neuroimaging, and molecular approaches to ensure timely and accurate diagnosis, as well as to inform long-term management strategies in affected individuals.

## Figures and Tables

**Figure 1 genes-16-01229-f001:**
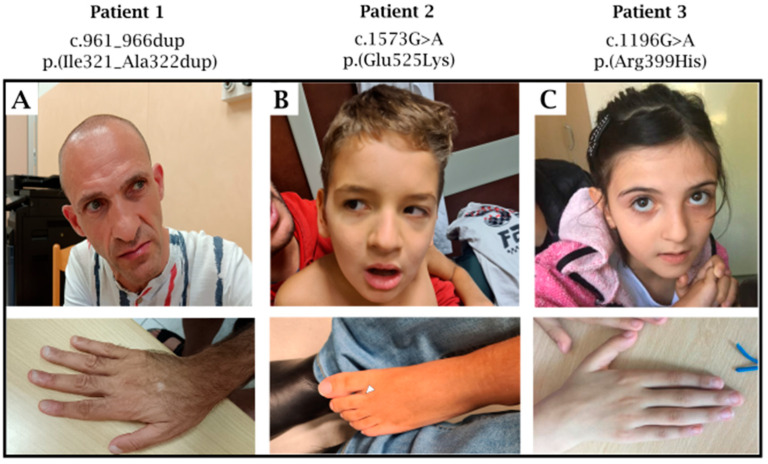
Patients 1–3 demonstrate nonspecific dysmorphic facial features such as: deep-set eyes, abnormal chin morphology, thin vermilion of the upper lip, long/smooth philtrum. (**A**) Patient 1 has a bilateral broad thumb; (**B**) Patient 2 shows bilateral partial proximal syndactyly of II_nd_ and III_rd_ foot fingers (arrowhead); (**C**) Patient 3 does not demonstrate a specific alteration in hand morphology. Individual ID, cDNA and protein effect are noted for each.

**Figure 2 genes-16-01229-f002:**
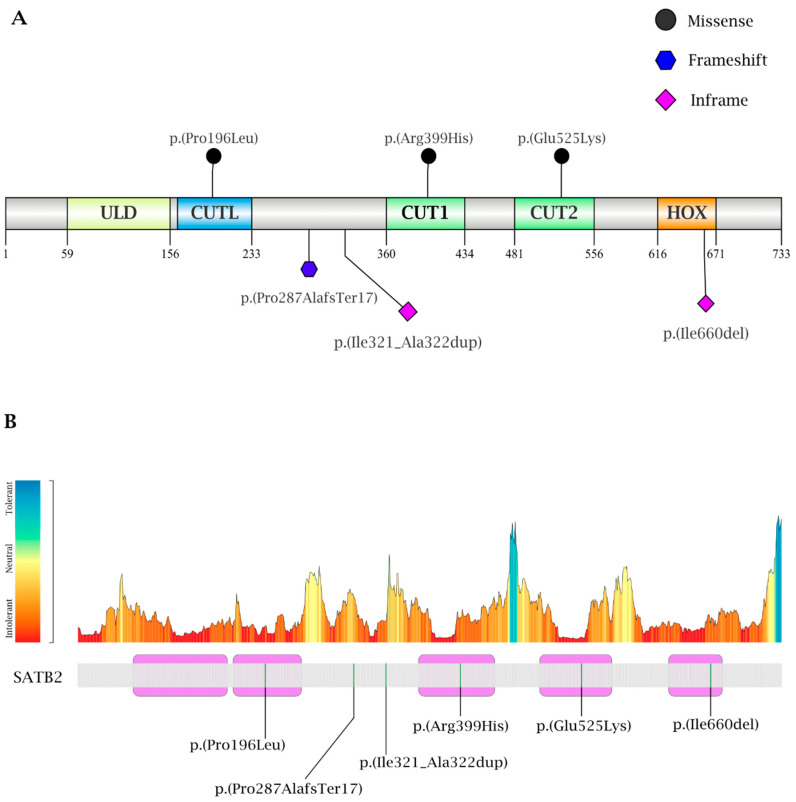
Distribution of identified variants along the SATB2 protein. (**A**) Variants are depicted at the protein level (p.) according to mutation type: black circles for missense variants, a blue hexagon for the frameshift variant, and a purple rhombus for in-frame variants. Diagrams were generated using Illustrator for Biological Sequences (IBS 2.0) [14]; (**B**) Graphical representation of regional intolerance to variation in SATB2 using MetaDome (https://stuart.radboudumc.nl/metadome; accessed on 1 October 2025). Variants identified in our cohort are mapped onto the protein structure. Purple segments denote the major functional domains: ULD, CUTL, CUT1, CUT2, and HOX. Affected amino acid residues are highlighted in green. All variants occur in regions predicted to be intolerant to variation, supporting their potential pathogenicity.

**Figure 3 genes-16-01229-f003:**
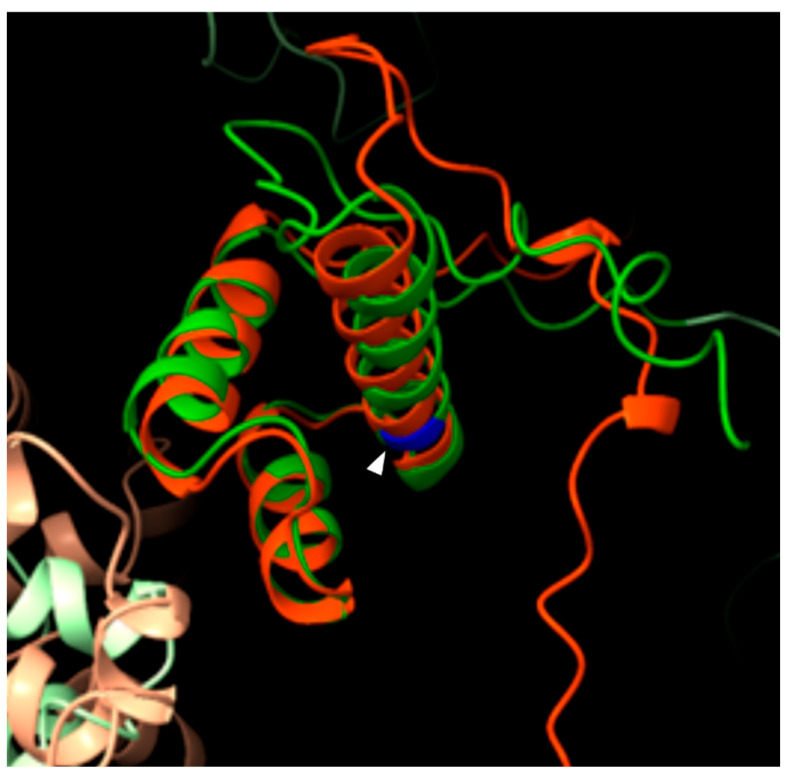
In silico protein folding modeling. Superimposition of SATB2 WT (green) and MUT (orange). White arrowhead indicates the position of the variant p.(Ile660del) (blue).

**Figure 4 genes-16-01229-f004:**
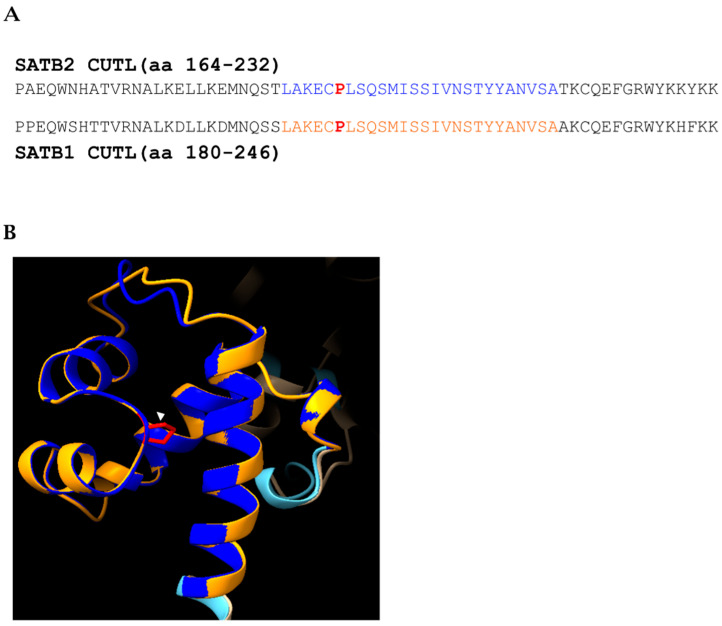
Sequence alignment of the SATB2 and SATB1 CUTL domains. (**A**) The overall sequence identity is 86%. Regions of 100% identity are highlighted in blue (SATB2) and orange (SATB1). The proline residue is shown in bold red, corresponding to p.Pro196 in SATB2 and p.Pro210 in SATB1. (**B**) Three-dimensional structural superimposition of the SATB2 (blue) and SATB1 (orange) CUTL domains. The proline residue at position 196 in SATB2 is highlighted in red and indicated by a white arrowhead.

**Table 2 genes-16-01229-t002:** *SATB2* variants interpretation of the six reported patients.

Case	cDNA	Protein	Inheritance	gnomAD	MetaRNN	MetaDome	Pat. Int.	ACMG
1	c.961_966dup	p.(Ile321_Ala322dup)	Unknown	Not found	N/A	Intolerant	IV—Likely Pathogenic	PM2 mod, PM4 mod, PP4 mod
2	c.1573G>A	p.(Glu525Lys)	De novo	Not found	0.75 (Supporting Pathogenic)	Highly intolerant	IV—Likely Pathogenic	PM1 mod, PP2 supp, PM2 mod, PS2 mod
3	c.1196G>A	p.(Arg399His)	De novo	Not found	0.85 (Moderate Pathogenic)	Intolerant	IV—Likely Pathogenic	PM5 mod, PM2 mod, PM1 mod, PP2 supp, PS2 mod
4	c.587C>T	p.(Pro196Leu)	De novo	Not found	0.86 (Moderate Pathogenic)	Intolerant	IV—Likely Pathogenic	PM2 mod, PP2 supp, PP3 supp, PS2 mod
5	c.857dup	p.(Pro287AlafsTer17)	De novo	Not found	N/A	N/A	V—Pathogenic	PVS1 very strong, PM2 mod, PS2 mod
6	c.1979_1981del	p.(Ile660del)	De novo	Not found	N/A	Intolerant	IV—Likely Pathogenic	PM2 mod, PM4 mod, PP5 supp, PS2 mod

Note: VUS, variant of unknown significance; mod, moderate; supp, supporting; N/A, not applicable; gnomAD: Genome Aggregation Database v.4.1.0; MetaRNN: pathogenicity prediction scores for human nonsynonymous SNVs (nsSNVs) http://www.liulab.science/metarnn.html; MetaDome: https://stuart.radboudumc.nl/metadome/ (accessed on 1 October 2025); Pat. Int.: Pathogenicity interpretation using ACMG classes; ACMG: detailed ACMG criteria.

## Data Availability

The original contributions presented in this study are included in the article/Appendix A. Further inquiries can be directed to the corresponding author.

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
