# Peer review of "Expanding Clinical and Genetic Landscape of SATB2-Associated Syndrome"

_genes, 2025, doi:10.3390/genes16101229_

Round 1
Reviewer 1 Report
Comments and Suggestions for Authors
The manuscript “Expanding clinical and genetic landscape of SATB2-associated Syndrome” by Pullano et al. reports six new cases of SATB2-associated syndrome and examines how each patient’s unique or rare SATB2 variant might influence their phenotype. The study includes the oldest known SAS patient, who is 56 years old. It features an unusual instance of cortical visual impairment, along with common characteristics such as speech disability, intellectual impairment, craniofacial and dental anomalies, and low body weight. The authors conduct variant–domain mapping and utilize multiple bioinformatic tools to support the pathogenicity of these variants. This case series provides a valuable contribution to the understanding of SAS; however, the analysis and interpretation would benefit from a more detailed mechanistic discussion and greater clarity. Specifically, the manuscript should strengthen the connection between molecular findings and biological outcomes to enhance conceptual coherence. Currently, it does not convincingly establish SAS as the pleiotropic outcome of a chromatin organizer’s partial loss, leading to disturbances in brain connectivity, craniofacial development, and energy homeostasis.
Major:
- The domain mapping provides a strong foundation; however, the discussion remains primarily descriptive. It is recommended that the authors elaborate on how modifications in CUTL, CUT1, CUT2, or HOX domains are expected to influence SATB2’s function, specifically regarding DNA binding, chromatin tethering, or interactions with partner proteins. Notably, variants in the CUT1/CUT2 domains are associated with more severe phenotypes and have been modeled to induce conformational changes. The authors should briefly delineate the biophysical or functional roles of these domains and discuss why perturbations could translate into larger neurodevelopmental effects. Additionally, the novel CUTL missense mutation p.Pro196Leu resides in a highly conserved region, yet the functional role of the CUT-like domain remains underexplored. The authors should include a figure based on SATB1 structural data to hypothesize whether the ULD–CUTL tandem coordinates DNA engagement, and whether substitutions in this region might impair chromatin docking or long-range genome organization.
- The HOX in-frame deletion warrants particular attention. The authors should include a concise, mechanistic hypothesis describing how a single-residue deletion near residue 660 could destabilize a homeodomain helix or disrupt DNA contact surfaces. This would elevate the schematic to a true mechanistic insight. Additionally, the authors should relate domain-level expectations to the individual cases. Specifically, if variants in DNA-engaging modules such as CUT1/CUT2/HOX are associated with more severe language or developmental deficits across the six cases, this relationship should be explicitly stated. Conversely, if the pattern is inconsistent, this should be acknowledged, with discussion of potential confounding factors.
- The authors should clarify the interpretation thresholds used in their analysis. While the manuscript states that all three missense variants are predicted to be deleterious by REVEL and CADD, two of these variants have REVEL scores around 0.36–0.39, which may not clearly indicate deleteriousness. The authors need to specify the cutoff thresholds for “deleterious" predictions in REVEL and differentiate between “supportive" and "decisive" evidence. Additionally, the manuscript should emphasize that high CADD scores and conservation metrics support potential pathogenicity but are not definitive on their own. These ensemble predictors are subject to false positives and negatives and should be interpreted in conjunction with de novo status and phenotypic fit. Since SATB2 is generally constrained across the genome, a lack of tolerated regions does not necessarily equate to functional intolerance without clear evidence. If there are regions within the gene that are tolerant, the authors should note that none of the variants in this cohort reside there; if not, they should clarify how they avoid over-interpreting constraint heatmaps.
- The authors should consider clarifying the use of the frameshift in a “slightly intolerant” region to underscore the point that gene-level haploinsufficiency can outweigh local tolerance, highlighting that regional maps, while informative, are not definitive indicators of truncation effects. Additionally, for clarity to a broader readership, it would be beneficial to briefly define REVEL and CADD as ensemble predictors, noting that higher scores are associated with an increased likelihood of deleteriousness. Including a sentence explaining this in the manuscript would enhance accessibility without compromising scientific rigor.
- The observations regarding cortical visual impairment (CVI) are compelling. The manuscript should more explicitly articulate the mechanistic link between SATB2’s role in specifying cortical neuron identity and inter-areal connectivity as a plausible pathway to CVI. Additionally, the authors should relate diffuse hypomyelination and white matter loss observed in the study to previous SAS findings. Since SATB2 influences callosal projection neuron development, its disruption could plausibly impair myelinated visual pathways and higher-order visual processing, thereby contributing to CVI. If CVI has not been previously reported in SAS, this should be clearly stated. Furthermore, cases of unexplained CVI with normal ocular examinations should prompt consideration of SAS as a potential underlying etiology.
- The MRI findings should be integrated into a cohesive neurodevelopmental framework. Evidence of corpus callosum hypoplasia, white matter anomalies, and language impairment could suggest a common theme related to cortical connectivity, consistent with SATB2 dysfunction. The authors should clearly delineate which aspects of their interpretations are supported by evidence and which are speculative. Employing concise, precise language that distinguishes observations from hypotheses will enhance the manuscript's clarity and credibility.
- The authors should substantiate their growth pattern observations, such as the prevalence of low weight with relatively preserved stature, by referencing published SAS data. Additionally, they should clarify whether their case studies represent typical trajectories or atypical variations. It would also strengthen the manuscript to consider the possibility that a single chromatin organizer misregulating gene networks across the hypothalamus, adipose tissue, or muscle could account for the observed growth and energy phenotypes.
Minor:
- In section 3.2.1, please rephrase the sentence to ensure that each missense variant is explicitly paired with its corresponding domain to prevent potential confusion.
The manuscript would benefit from the inclusion of domain-specific mechanisms, clearer delineation of the limits of in silico inference, a more precise articulation of neurodevelopmental and metabolic logic, and better integration of evidence across scales. These enhancements would improve its cohesiveness and strengthen its conceptual contribution.
Author Response
The manuscript “Expanding clinical and genetic landscape of SATB2-associated Syndrome” by Pullano et al. reports six new cases of SATB2-associated syndrome and examines how each patient’s unique or rare SATB2 variant might influence their phenotype. The study includes the oldest known SAS patient, who is 56 years old. It features an unusual instance of cortical visual impairment, along with common characteristics such as speech disability, intellectual impairment, craniofacial and dental anomalies, and low body weight. The authors conduct variant–domain mapping and utilize multiple bioinformatic tools to support the pathogenicity of these variants. This case series provides a valuable contribution to the understanding of SAS; however, the analysis and interpretation would benefit from a more detailed mechanistic discussion and greater clarity. Specifically, the manuscript should strengthen the connection between molecular findings and biological outcomes to enhance conceptual coherence. Currently, it does not convincingly establish SAS as the pleiotropic outcome of a chromatin organizer’s partial loss, leading to disturbances in brain connectivity, craniofacial development, and energy homeostasis.
Major:
- The domain mapping provides a strong foundation; however, the discussion remains primarily descriptive. It is recommended that the authors elaborate on how modifications in CUTL, CUT1, CUT2, or HOX domains are expected to influence SATB2’s function, specifically regarding DNA binding, chromatin tethering, or interactions with partner proteins. Notably, variants in the CUT1/CUT2 domains are associated with more severe phenotypes and have been modeled to induce conformational changes. The authors should briefly delineate the biophysical or functional roles of these domains and discuss why perturbations could translate into larger neurodevelopmental effects. Additionally, the novel CUTL missense mutation p.Pro196Leu resides in a highly conserved region, yet the functional role of the CUT-like domain remains underexplored. The authors should include a figure based on SATB1 structural data to hypothesize whether the ULD–CUTL tandem coordinates DNA engagement, and whether substitutions in this region might impair chromatin docking or long-range genome organization.
Answer: In discussion, we extended the description of SATB2 functional domains, and discussed the correlation between CUT1/CUT2 variants and the clinical presentation in SAS patients. We confirm that also our mutations in these domains are associated with a more severe phenotype as described in the literature.
As requested, we have included Figure 4, which presents both the sequence alignment and the three-dimensional structural comparison of the CUTL domain between SATB1 and SATB2.
We discussed the role of our variants in the context of the high level of homology with the well functionally characterized SATB1 ULD-CUTL tandem domain. In discussion we added “However structural and biochemical work on SATB1 demonstrated that the ULD–CUTL tandem is essential for the multiple-domain–coordinated mechanism of DNA recognition, with CUTL contributing, beyond CUT1, CUT2 and HOX, to the DNA binding capacity of full-length protein and enabling simultaneous engagement of two DNA targets, a plausible basis for MAR-anchored loop formation. The high degree of homology (86%) between the CUTL domains of SATB1 and SATB2, reaching complete conservation at the position of the missense variant p.(Pro196Leu) (Figure 4A,B), suggests a potential effect on ULD–CUTL coupling and oligomerization, ultimately leading to reduced multi-site DNA engagement and defective chromatin-loop organization as a potential pathogenic mechanism [23]”.
- The HOX in-frame deletion warrants particular attention. The authors should include a concise, mechanistic hypothesis describing how a single-residue deletion near residue 660 could destabilize a homeodomain helix or disrupt DNA contact surfaces. This would elevate the schematic to a true mechanistic insight. Additionally, the authors should relate domain-level expectations to the individual cases. Specifically, if variants in DNA-engaging modules such as CUT1/CUT2/HOX are associated with more severe language or developmental deficits across the six cases, this relationship should be explicitly stated. Conversely, if the pattern is inconsistent, this should be acknowledged, with discussion of potential confounding factors.
Answer: We appreciate the observations. In the results section, we added Figure 3, which shows the 3D protein folding model of the SATB2 wild-type and mutant HOX domain, superimposed using AlphaFold. Stability prediction analysis was added to highlight the possible role of p.(Ile660del) variant in the loss of functional integrity of the HOX domain as a pathogenic mechanism that led to a SAS phenotype.
We added: “Exploiting in silico 3D protein stability prediction analysis, we generated and compared the WT and mutated (p.(Ile660del)) form of the SATB2 HOX domain (figure 3). The prediction shows that apparently there is a gain in protein stability (ΔΔG= −51.18 kcal/mol), suggesting an increased rigidity of this region suggesting a possible impact on the protein function that need more functional evidences (Figure 3).”
Moreover, in discussion we added: “Another novel variant, p.(Ile660del), was found in patient 6, affecting the HOX domain. Recently, functional assays have shown that missense variants within this domain impair transcriptional repression and disrupt normal nuclear localization, resembling the effects observed in HOX-domain-lacking truncating variants and supporting its critical role in MAR (Matrix Attachment Region) binding and chromatin remodeling activity. It is therefore conceivable that the variant p.(Ile660del) has a similar role in the loss of functional integrity of the HOX domain [24]. In addition, 3D folding-based protein stability predictions indicate that the mutant protein is more stable than the wild type, suggesting that altered protein stability may represent a possible pathogenic mechanism (Figure 3).”
- The authors should clarify the interpretation thresholds used in their analysis. While the manuscript states that all three missense variants are predicted to be deleterious by REVEL and CADD, two of these variants have REVEL scores around 0.36–0.39, which may not clearly indicate deleteriousness. The authors need to specify the cutoff thresholds for “deleterious" predictions in REVEL and differentiate between “supportive" and "decisive" evidence. Additionally, the manuscript should emphasize that high CADD scores and conservation metrics support potential pathogenicity but are not definitive on their own. These ensemble predictors are subject to false positives and negatives and should be interpreted in conjunction with de novo status and phenotypic fit. Since SATB2 is generally constrained across the genome, a lack of tolerated regions does not necessarily equate to functional intolerance without clear evidence. If there are regions within the gene that are tolerant, the authors should note that none of the variants in this cohort reside there; if not, they should clarify how they avoid over-interpreting constraint heatmaps.
Answer: We thank the reviewer for raising this important point. We agree that bioinformatics predictors can sometimes yield discordant results and are not always easy to interpret. To address this, we revised our approach and now use the MetaRNN score (http://www.liulab.science/metarnn.html), which integrates information from 28 high-level annotation scores and provides a final prediction representing the likelihood that a non-synonymous single nucleotide variant (nsSNV) is pathogenic. We have updated the text accordingly to emphasize that the computationally derived assessment of missense variant deleteriousness is suggestive but not conclusive.
- The authors should consider clarifying the use of the frameshift in a “slightly intolerant” region to underscore the point that gene-level haploinsufficiency can outweigh local tolerance, highlighting that regional maps, while informative, are not definitive indicators of truncation effects. Additionally, for clarity to a broader readership, it would be beneficial to briefly define REVEL and CADD as ensemble predictors, noting that higher scores are associated with an increased likelihood of deleteriousness. Including a sentence explaining this in the manuscript would enhance accessibility without compromising scientific rigor.
Answer: We thank the reviewer for highlighting this point, which we consider a mistake. The frameshift variant is indeed likely to cause haploinsufficiency. We changed the part relative to bioinformatic predictors to avoid misinterpretation of the use of multiple predictors. Changed also figure 2B, removing the indication of the frameshift variant from Metadome intolerance map.
- The observations regarding cortical visual impairment (CVI) are compelling. The manuscript should more explicitly articulate the mechanistic link between SATB2’s role in specifying cortical neuron identity and inter-areal connectivity as a plausible pathway to CVI. Additionally, the authors should relate diffuse hypomyelination and white matter loss observed in the study to previous SAS findings. Since SATB2 influences callosal projection neuron development, its disruption could plausibly impair myelinated visual pathways and higher-order visual processing, thereby contributing to CVI. If CVI has not been previously reported in SAS, this should be clearly stated. Furthermore, cases of unexplained CVI with normal ocular examinations should prompt consideration of SAS as a potential underlying etiology.
Answer: We appreciated the reviewer’s punctual observation. We expanded the discussion by implementing evidence of SATB2 expression specifically in the occipital cortex derived from Protein Atlas and highlighting the role of SATB2 in the regulation of the callosal projection during cerebral development.
- The MRI findings should be integrated into a cohesive neurodevelopmental framework. Evidence of corpus callosum hypoplasia, white matter anomalies, and language impairment could suggest a common theme related to cortical connectivity, consistent with SATB2 dysfunction. The authors should clearly delineate which aspects of their interpretations are supported by evidence and which are speculative. Employing concise, precise language that distinguishes observations from hypotheses will enhance the manuscript's clarity and credibility.
Answer: We improved and integrated into neurodevelopmental SATB2 features the neuroimaging description in discussion as follows: “Moreover, in patients 2 and 6 neuroradiological anomalies were reported as diffuse white matter hypomyelination and volume loss for patient 2 and hypoplasia of the corpus callosum for patient 6. Those findings are consistent with previous reports that describe abnormality of white matter, delay of myelination, ventriculomegaly and small corpus callosum as the four most frequently reported neuroimaging anomalies [25].”
- The authors should substantiate their growth pattern observations, such as the prevalence of low weight with relatively preserved stature, by referencing published SAS data. Additionally, they should clarify whether their case studies represent typical trajectories or atypical variations. It would also strengthen the manuscript to consider the possibility that a single chromatin organizer misregulating gene networks across the hypothalamus, adipose tissue, or muscle could account for the observed growth and energy phenotypes.
Answer: Unfortunately, the phenotype of our patients lacks a detailed auxological characterization. We could only identify the prevalence of low weight with relatively preserved stature data was retrieved from punctual clinical observation and family reports. We provide a more detailed reference to already published data on growth in SAS patients and the possible role of SATB2 in causing a misregulation in energy-related molecular networks.
Minor:
- In section 3.2.1, please rephrase the sentence to ensure that each missense variant is explicitly paired with its corresponding domain to prevent potential confusion.
Answer: We agree with the punctual suggestion, and we reformulated the sentence to obtain a clearer exposition of the results.
Reviewer 2 Report
Comments and Suggestions for Authors
The manuscript entitled "Expanding clinical and genetic landscape of SATB2-associated Syndrome" addresses a clinically significant and underdiagnosed neurodevelopmental disorder, SATB2-associated syndrome (SAS), and reports six new cases with rare or novel SATB2 variants. The paper is well written and logically structured, and provides new insights, including new clinical observations such as a case of adult visual impairment and cortical disorders. The bioinformatic analyzes are appropriate, and the clinical discussion is relevant and timely. I have only minor comments and suggestions for improvement.
1) Consider repositioning Figure 1 so it appears closer to the clinical descriptions of Patients 1–3, which would improve readability.
2) The in-frame duplication in Patient 1 is currently classified as VUS, but given the high conservation and phenotypic match, the authors might consider elaborating on the potential pathogenicity of this variant and its relevance for future reclassification. This would emphasize its clinical relevance despite the current VUS classification.
3) The finding of cortical visual impairment in Patient 2 is unusual and interesting in the context of SAS. The authors already highlight the cortical origin, but they could expand the discussion by exploring whether SATB2 has a known role in visual cortical development, myelination, or related neurodevelopmental processes, based on previous studies.
4) While the manuscript discusses overall phenotypic variability, it would be useful to clearly list which specific clinical features are newly described or particularly rare compared to previous SAS cohorts (e.g., cortical visual loss, metabolic findings, toe syndactyly, etc.).
5) Minor typos should be corrected. “STAB2” instead of “SATB2” (page 7, line 257). The authors should ensure consistent nomenclature (SATB2 in italics for the gene, not in italics for the protein). Also, some inconsistencies in formatting, e.g. incorrect spacing in column headings (e.g., “Languag e delay” in Table 1).
Author Response
The manuscript entitled "Expanding clinical and genetic landscape of SATB2-associated Syndrome" addresses a clinically significant and underdiagnosed neurodevelopmental disorder, SATB2-associated syndrome (SAS), and reports six new cases with rare or novel SATB2 variants. The paper is well written and logically structured, and provides new insights, including new clinical observations such as a case of adult visual impairment and cortical disorders. The bioinformatic analyzes are appropriate, and the clinical discussion is relevant and timely. I have only minor comments and suggestions for improvement.
- Consider repositioning Figure 1 so it appears closer to the clinical descriptions of Patients 1–3, which would improve readability.
Answer: We agree with the punctual suggestion. The figure has been repositioned as suggested.
- The in-frame duplication in Patient 1 is currently classified as VUS, but given the high conservation and phenotypic match, the authors might consider elaborating on the potential pathogenicity of this variant and its relevance for future reclassification. This would emphasize its clinical relevance despite the current VUS classification.
Answer: We thank the reviewer for this observation and agree that the phenotype was concordant and specific for SAS. We accepted the suggestion to add the PP4 criterium, and change the ACMG classification to LP.
- The finding of cortical visual impairment in Patient 2 is unusual and interesting in the context of SAS. The authors already highlight the cortical origin, but they could expand the discussion by exploring whether SATB2 has a known role in visual cortical development, myelination, or related neurodevelopmental processes, based on previous studies.
Answer: We expanded the discussion by implementing evidence of SATB2 expression specifically in the occipital cortex derived from ProteinAtlas and highlighting the role of SATB2 in the regulation of the callosal projection during cerebral development.
In discussion, we added: “Interestingly patient 2 presented with a unique visual phenotype of severely reduced visual acuity without apparent ocular abnormalities combined with neuroradiological features of diffuse white matter hypomyelination and volume loss, suggesting a cortical origin of the vision loss. It is the first time, to our knowledge, that a central vision impairment is reported in a SAS patient. According to the Human Protein Atlas, SATB2 shows strong nuclear expression in pyramidal neurons of the cerebral cortex, including the occipital lobe and visual areas supporting its potential role in the development and function of cortical circuits essential for the processing of visual information combined with the assessed role of SATB2 the regulation of the callosal projection during cerebral development [26]”.
- While the manuscript discusses overall phenotypic variability, it would be useful to clearly list which specific clinical features are newly described or particularly rare compared to previous SAS cohorts (e.g., cortical visual loss, metabolic findings, toe syndactyly, etc.).
Answer: The only specific clinical feature we identified is the visual phenotype of severely reduced visual acuity without apparent ocular abnormalities in patient 2. All other clinical features, given the clinical variability highlighted were already reported. We cited this point in the abstract and conclusion. We must also be careful to associate this feature to SATB2. We think this observation can be confirmed in future SAS cases.
- Minor typos should be corrected. “STAB2” instead of “SATB2” (page 7, line 257). The authors should ensure consistent nomenclature (SATB2 in italics for the gene, not in italics for the protein). Also, some inconsistencies in formatting, e.g. incorrect spacing in column headings (e.g., "Language delay” in Table 1).
Answer: We thank reviewer for highlighting these points that we carefully edited in the text.
Round 2
Reviewer 1 Report
Comments and Suggestions for Authors
The revised manuscript "Expanding the clinical and genetic landscape of SATB2-associated syndrome" by Pullano et al. improves the domain-level narrative, strengthens the HOX domain in-silico analysis, and adds a useful CUTL alignment with structural overlay that provides a mechanistic context. The shift from REVEL and CADD to MetaRNN correctly redefines predictor evidence as supportive rather than decisive. The discussion of cortical visual impairment and white matter or callosal findings is more explicit. Several concerns still need attention.
- The manuscript should present genotype–phenotype correlations in a structured format. Provide a supplemental CSV with a severity matrix that has domain classes on one axis and features on the other. Include expressive language, global developmental delay, MRI findings of callosal and white matter abnormalities, and dental or palatal anomalies. Encode each feature with Human Phenotype Ontology terms. Add Zarate SAS severity scores for each case. Perform Fisher’s exact tests for binary traits and rank-based tests for ordinal traits, and report effect directions with 95 percent confidence intervals. Summarize the findings clearly with plots or a heatmap. This converts the current narrative into an auditable dataset.
- The manuscript describes p.Ile660del as novel and states that it has been reported in ClinVar as pathogenic. These claims contradict each other. If the authors have not contributed to this ClinVar entry, remove the novelty claim, cite the ClinVar record with its accession and access date, and frame the contribution as an additional case with phenotype and modeling context. This correction ensures accuracy in variant interpretation.
- The evidence for p.Ile321_Ala322dup is insufficient for a likely pathogenic classification. Authors should provide trio Sanger confirmation with parentage, report segregation or LOD data where available, and demonstrate absence in gnomAD with ancestry-matched checks. The manuscript text should clarify the biological relevance within the implicated domain to justify PM4 and, if appropriate, PM1. Define PP4 by listing the exact SAS-defining HPO terms. Keep the variant label consistent across text and figures. Include a per-variant ACMG worksheet in the Supplement. If these data are unavailable, reclassify the variant as VUS, aligning with standard guidelines.
- Authors should expand the modeling methods. Specify the pipeline and include per-residue pLDDT for the affected region. Report the FoldX version, the number of replicates, and the mean and standard deviation of delta-delta-G with a clear sign convention indicating mutant minus wild type. Add a local structural superposition around the altered site to show any register shifts or changes in DNA contacts. State the error bounds of AlphaFold and FoldX, and treat delta-delta-G as qualitative until functional assays support its interpretation.
- The predictor evidence handling in the manuscript needs strengthening. Authors should add a Methods section mapping MetaRNN score ranges to evidence strengths and consistently use the phrase supportive but non-conclusive. Remove PP5 unless its provenance justifies it at the stated strength. Correct any duplicated or mistyped ACMG codes. Provide a supplementary table listing, for each variant, the applied ACMG criterion, the specific evidence, and a citation.
The study shows promise, but these issues must be addressed before conclusions can be accepted. With these revisions, the manuscript will meet the necessary standards of rigor and reproducibility for clinical variant interpretation.
Author Response
We gratefully thank reviewer for her/his thoughtful suggestions. Below the point-by point replies.
1) The manuscript should present genotype–phenotype correlations in a structured format. Provide a supplemental CSV with a severity matrix that has domain classes on one axis and features on the other. Include expressive language, global developmental delay, MRI findings of callosal and white matter abnormalities, and dental or palatal anomalies. Encode each feature with Human Phenotype Ontology terms. Add Zarate SAS severity scores for each case. Perform Fisher’s exact tests for binary traits and rank-based tests for ordinal traits, and report effect directions with 95 percent confidence intervals. Summarize the findings clearly with plots or a heatmap. This converts the current narrative into an auditable dataset.
Answer:We realized a supplemental phenotype-genotype correlation CSV file that includes:
- Domain, cDNA, Protein, Origin, CP, Low BMD, Abnl MRI, Walk at (months), Talk at (months), Speech (words), Abnormal Teeth, Seizures, Abnormal behaviour, Abnormal sleep, Severity score: Neurodevelopment total, Severity score: Systemic total, Severity score: total) using the satb2-portal (https://satb2-portal.broadinstitute.org/) reference CSV (Table S3).
- Heatmap graphical representation of single Zarate SAS severity scores features indexed using Human Phenotype Ontology (HPO) as reference. (Table S2).
- We considered the reviewer’s suggestion to perform statistical analyses to evaluate potential differences among previously reported cases; however, given the single representation within each domain, we opted not to conduct such analyses due to the limited statistical power in this context.
2) The manuscript describes p.Ile660del as novel and states that it has been reported in ClinVar as pathogenic. These claims contradict each other. If the authors have not contributed to this ClinVar entry, remove the novelty claim, cite the ClinVar record with its accession and access date, and frame the contribution as an additional case with phenotype and modeling context. This correction ensures accuracy in variant interpretation.
Answer: We thank the reviewer for the comment, that was indeed a mistake. We removed the word “novel” in the discussion when referring to this variant (at the beginning of the paragraph after figure 4).
We already reported in the paragraph “Indel Variants” the Clinvar ID of this variant.
We added Table S4 adding the Clinvar information for all the variants.
3) The evidence for p.Ile321_Ala322dup is insufficient for a likely pathogenic classification. Authors should provide trio Sanger confirmation with parentage, report segregation or LOD data where available, and demonstrate absence in gnomAD with ancestry-matched checks. The manuscript text should clarify the biological relevance within the implicated domain to justify PM4 and, if appropriate, PM1. Define PP4 by listing the exact SAS-defining HPO terms. Keep the variant label consistent across text and figures. Include a per-variant ACMG worksheet in the Supplement. If these data are unavailable, reclassify the variant as VUS, aligning with standard guidelines.
Answer: As stated in the manuscript (description of Patient 1), parents were unavailable for segregation analysis, therefore Sanger sequencing could only be performed in patient. Even in the absence of a de novo variant, ACMG classification of the p.(Ile321_Ala322dup) can lead to a likely pathogenic result.
The ACMG criteria used were:
- PM2: The variant is absent from gnomAD v4.1 (taking into account 807.162 samples).
- PM4: applied as moderate because it is an in-frame duplication in a non-repetitive region. Analysis of the region where the amino acid is inserted shows that it is highly conserved and located near the beginning of the CUT1 domain.
- PP4 moderate: The exact SAS-defining HPO that correspond to our patient’s phenotype are:
Intellectual disability HP:0001249, Delayed speech and language development HP:0000750, Expressive language delay HP:0002474, Atypical behavior HP:0000708, Sleep abnormality HP:0002360, Scoliosis HP:0002650, Abnormal muscle tone HP:0003808, Growth abnormality HP:0001507, Abnormality of the dentition HP:0000164.
With a total score of 23 (based on SATB2 portal (Table S2,S3)), we applied PP4 as moderate because the patient’s phenotype was specific for SAS.
We edited variants nomenclature in the overall manuscript together with figure 2A to make it more consistent. We thank the reviewer for highlighting this point.
4) Authors should expand the modeling methods. Specify the pipeline and include per-residue pLDDT for the affected region. Report the FoldX version, the number of replicates, and the mean and standard deviation of delta-delta-G with a clear sign convention indicating mutant minus wild type. Add a local structural superposition around the altered site to show any register shifts or changes in DNA contacts. State the error bounds of AlphaFold and FoldX, and treat delta-delta-G as qualitative until functional assays support its interpretation.
Answer: we provided an expanded version of the modeling methods as requested and added a supplementary file with extended modeling data.
“Modeling pipeline and statistics. Wild-type (WT) structures were retrieved from the AlphaFold Protein Structure Database [15], and variant backbones were modeled with the AlphaFold server from the corresponding mutated amino-acid sequences. Resulting mmCIFs were converted to PDB and pre-optimized with FoldX v5 (RepairPDB, 3 consecutive passes per model). We then ran Stability independently on WT and mutant models. For each variant, ΔΔG was defined as TotalEnergy(Mutant) − TotalEnergy(WT) (kcal/mol; ΔΔG > 0 = destabilizing; ΔΔG < 0 = stabilizing). To account for model uncertainty, ΔΔG was computed across the five AlphaFold ranked models per genotype (ranked_0–ranked_4) and summarized as mean ± SD with 95% confidence intervals.
Confidence and visualization. Per-residue confidence (pLDDT) was read from the B-factor field of AlphaFold PDBs; for the affected region (SATB2 HOX domain residues 616–671, ±10 residues) we report the per-residue values and their mean ± SD (Table S1), and we interpret geometry primarily where pLDDT ≥ 70. Local WT–mutant superpositions were performed in UCSF ChimeraX [v1.10.1] over Cα atoms within the 616–671 residue window around the altered site.
Error bounds. AlphaFold confidence limits (pLDDT/PAE) and the empirical nature of FoldX imply typical uncertainties. Accordingly, ΔΔG values are treated as qualitative indicators until supported by functional assays.”
We think that this work suggests a change in protein structure. However, we think a dynamic modelling showing the protein-DNA interaction is beyond the scope of this manuscript.
5) The predictor evidence handling in the manuscript needs strengthening. Authors should add a Methods section mapping MetaRNN score ranges to evidence strengths and consistently use the phrase supportive but non-conclusive. Remove PP5 unless its provenance justifies it at the stated strength.
Answer:We strengthen the method section adding the score ranges applied to MetaRNN prediction analysis for our variants.
As reported in the materials and methods section:
“Bioinformatic prediction for missense variants were evaluated using MetaRNN (http://www.liulab.science/metarnn.html), a pathogenicity prediction score for human nonsynonymous SNVs (nsSNVs). It combined data from 28 high-level annotation scores, comprising 16 functional prediction scores (like SIFT, PolyPhen2_HDIV, PolyPhen2_HVAR, MutationAssessor, PROVEAN, VEST4, M-CAP, REVEL, MutPred, MVP, PrimateAI, DEOGEN2, CADD, fathmm-XF, Eigen, and GenoCanyon), 8 conservation scores (GERP, phyloP100way_vertebrate, phyloP30way_mammalian, phyloP17way_primate, phastCons100way_vertebrate, phastCons30way_mammalian, phastCons17way_primate, and SiPhy), and 4 allele frequency datasets (1000 Genomes Project, ExAC, gnomAD exome, and gnomAD genome).
These inputs are integrated into an ensemble prediction framework based on a deep recurrent neural network (RNN), which outputs the probability of a non-synonymous single nucleotide variant (nsSNV) being pathogenic.
MetaRNN outputs a probability score ranging from 0 to 1, with values >0.5 generally indicating predicted pathogenicity [12].
In accordance with a recently proposed calibration strategy (Cristofoli et al., MAGI-ACMG, Genes 2023, PMID: 37628650), we adopted the following thresholds to map MetaRNN scores to evidence strengths (≥0.748: supporting evidence of pathogenicity, ≥0.841: moderate evidence of pathogenicity, ≥0.939: strong evidence of pathogenicity). However, in line with ACMG/AMP guidelines, all in silico predictors were consistently considered as supportive but non-conclusive evidence. No variant was classified solely on the basis of predictor scores.”
Following the reviewer suggestion, we changed the PP5 strength to Supporting.
6) Correct any duplicated or mistyped ACMG codes. Provide a supplementary table listing, for each variant, the applied ACMG criterion, the specific evidence, and a citation.
Answer: We edited any duplicated or mistyped ACMG criteria and provided a supplementary Table S4 with all the information required.